# An Innovative Comparative Analysis Approach for the Assessment of Laparoscopic Surgical Skills

**Saiteja Malisetty** [1,*] , **Hesham H. Ali** [1], **Elham Rastegari** [2] **and Ka-Chun Siu** [3]

1    College of Information Science & Technology, University of Nebraska at Omaha, Omaha, NE 68182, USA
2    Business Intelligence & Analytics Department, Creighton University, Omaha, NE 68178, USA
3    Department of Health & Rehabilitation Sciences, University of Nebraska Medical Center, Omaha, NE 68198, USA
*    Correspondence: smalisetty@unomaha.edu; Tel.: +1-(402)-807-9060

**Abstract:** Over the past few decades, surgeon training has changed dramatically. Surgical skills are now taught in a surgical skills laboratory instead of the operating room. Simulation-based training helps medical students improve their skills, but it has not revolutionized clinical education. One critical barrier to reaching such a desired goal is the lack of reliable, robust, and objective methods for assessing the effectiveness of training sessions and the development of students. In this paper, we will develop a new comparative analysis approach that employs network models as the central concept in establishing a new assessment tool for the evaluation of the surgical skills of trainees as well as the training processes. The model is populated using participants electromyography data while performing a simulation task. Furthermore, using NASA Task Load Index score, participants' subjective overload levels are analyzed to examine the impact of participants' perception of their mental demand, physical demand, temporal demand, performance, effort, and frustration on how participants perform each simulation task. Obtained results indicate that the proposed approach enables us to extract useful information from the raw data and provides an objective method for assessment the of surgical simulation tasks and how the participants' perception of task impacts their performance.

**Keywords:** laparoscopic surgery; simulation; network models; enrichment analysis; NASA-TLX



## 1. Introduction

There is a growing shortage of medical professionals all around the world. By 2030, it is predicted that there will be a global shortage of 10 million healthcare workers, mainly in low- and middle-income nations. In low- and middle-income nations, an additional 143 million surgical procedures are required annually to save lives and prevent disability [1]. Unfortunately, even developed nations such as the United States are not producing enough new medical professionals fast enough to keep up with its aging population and falling birth rate [1]. Obtaining surgical competence is a difficult procedure requiring the acquisition of information, judgment, professionalism, and surgical skill. Since the beginning of the last century, doctors have been educated using the Halstedian model of surgical training, which entails learning the skill of surgery through an apprenticeship [2]. Although this model of surgical education has been successful in the past in producing a qualified surgical workforce, a multiplicity of circumstances has influenced the need to reevaluate pedagogical tactics in surgical education [3]. One inefficient aspect of current training methods is having the trainer sit down with the learner and play back recordings of the procedures being taught in order to get them up to speed on the necessary techniques.

Simulation is an educational method that allows the learner to operate interactively in an environment that recreates or replicates a clinical scenario from the real world but is not identical to "real life" [4]. Perhaps one of the most compelling motivations for the incorporation of simulation into surgical training is the ethical duty to provide the best care

possible to patients. Although it is accepted that trainees will eventually learn technical skills by treating patients, patients should not be subjected to the risk of damage when other ways of skill acquisition are available. Before trainees treat real patients, simulation guarantees that they have had some practice. Simulation also enables other methods of skill acquisition within the limits of work-hour restrictions and limited clinical experience [5].

Laparoscopy, a common type of minimally invasive surgery, is performed through one or more small incisions using small tubes, miniature video cameras, and surgical equipment [6]. In the field of laparoscopic surgery, minimally invasive surgical devices are becoming more prevalent and complex. Doctors must learn how to use modern technologies efficiently to remain updated within the current medical practice. However, it is difficult to determine the efficiency of the doctors while using the new devices. There are no quantitative assessment standards to evaluate users' performance with the new devices. Some individualized performance-based methods are utilized in the literature such as the Objective Structured Assessment of Technical Skills (OSATS) [7], Observational Clinical Human Reliability Assessment (OCHRA) [8], and hand motion analysis [9,10].

OSATS is a surgical skill assessment method that can be used to quantify a trainee's surgical skills. However, in order to get surgeons to rate a trainee's performance, surgeons themselves have to observe the trainee's performance, which is labor expensive [7]. OCHRA is an evaluation method based on recorded footage of an operation and distinguishes the surgeons who made the most and the least mistakes [8]. This approach assesses the total skills of advanced laparoscopic surgery, considering the amount of tissue handling procedures, instrument misuse, and consequential mistakes, and does not directly analyze the patterns of hand movement of the individuals to assess the operation itself with greater precision. Using Machine Learning algorithms (Logistic Regression & Support Vector Machines) to group surgeons with various levels of experience has yielded an extremely low accuracy [11–13]. Furthermore, when classifying surgeons using Machine Learning techniques, you lack the pedigree to notice subgroups that have evolved between subjects within the same classification group.

Hence, we believe that there is still a lack of a sophisticated and robust model that could assess individuals' skill levels by utilizing a portable simulation trainer and optimize the training schedules based on the performance of the subjects. Graph theory is the study of graphs and mathematical structures that model the relationships between objects. It is widely used to find the communities or clusters, optimal paths, and group correlations in many application domains [14]. Recent advances in network models have the potential to greatly enhance the usefulness of training in simulation settings by enabling the simulation to assess the trainee's mastery on their own. To evaluate individuals in relation to those undergoing the same training, the suggested approach applied the idea of comparative population-based analysis [14]. Such context-rich evaluation permits us to evaluate not only the students themselves but also the training sessions. To determine which participants perform similarly to one another and which ones perform distinctively, we use the comparative analysis approach using the network model, which forms a graph using the correlation between the subjects based on their muscle movements. In this network model, the nodes represent the subjects, and the edges represent the similarity score which is determined using the correlation coefficient. This approach has been successfully applied to other application domains and was shown to have high degrees of accuracy and robustness [15]. In our study, this method uses the muscle features associated with different levels of trainees and differentiates experts from novices. We believe that these types of network models could be useful in creating an automatic and objective method for assessing individual performance at a cohort or population level, which will provide a more comprehensive and inclusive outlook on the change of performances.

When undertaking surgical training that needs more physical and mental stability, trainees appear to exhibit high levels of stress and uncertainty. Obtaining mental workload levels during task performance is a difficult procedure. The workload level experienced by a trainee can affect their performance in any task [16]. This effect can be caused by either

excessive or reduced mental workload. Thus, estimating workload levels can help isolate sources that affect performance. In the current study, we use the NASA Task Load Index (NASA-TLX), which is a subjective, multidimensional evaluation method where workloads received scores to determine the effectiveness or other aspects of the performance of a task, device, or team to study the factors that impact the performance of the medical trainees [17].

By utilizing this method of comparing the progress of participants to one another in an objective manner, an instructor could theoretically identify trainees who require more training than is typically provided, as well as trainees who are mastering the skills quicker than expected. These insights can then be used to create personalized training schedules, where additional guidance is offered for a particular skill for a particular group depending on the performance of each and his/her proficiency level for each task relative to the rest of their class. In the sections that follow, we present the activities involved in acquiring data, setting up an experiment, pre-processing the data using feature selection, and finally, developing a network model.

## 2. Materials and Methods

### 2.1. Subjects

Using a virtual training system developed by a local Medical Center [18], twenty participants (8 female and 12 male) were recruited from a variety of disciplines (2 medical fellows, 8 medical students, and 10 non-medical students). However, due to challenges with data collection, participants 19 and 20 were excluded from the analysis, as the data on their performances was incomplete. Of the eighteen participants included in the analysis, nine were from a medical background (medical student or fellow) and nine were from a non-medical background (students in kinesiology, physical therapy, nursing, or radiology). All of the subjects were recruited from the university campus, and none of them had any prior experience with the research training simulator or any recent upper arm injuries that would exclude them from using the simulator.

### 2.2. Tasks

All participants completed a set of basic simulation training tasks (peg transfer, needle passing, and wire loop), and for this report to demonstrate the use of the network model, the needle passing (NP) task was selected as shown in Figure 1. NP task involves a plate with 6 holes in it. Using two virtual laparoscopic instruments, the participant must guide a needle through the hole to the other side of the plate. They must then guide the needle into the next hole and pass it through again. The task is complete when the needle has been guided through all of the holes.

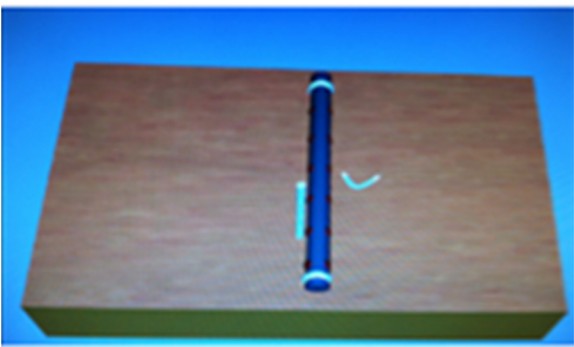

**Figure 1.** Needle passing task.

### 2.3. Experiment Setup

All participants were asked to engage in the experimental protocol during a 4-week period. This protocol included one pretraining test, three training sessions, and one post-training test. All participants performed the complete set of basic training tasks in all three

sessions. To get better precision in each session, each task is repeated five times by all participants. Data on participants' performance was collected in three sessions: an initial baseline session, a session one week after baseline, and a session four weeks after baseline. Upon finishing the task, each participant is asked to give feedback on their physical and mental overload levels based on the NASA TLX scores.

*2.4. Data Collection*

The study was conducted in accordance with the Declaration of Helsinki and approved by the Institutional Review Board of the University of Nebraska Medical Center (IRB # 103-12-EX) on 11 January 2017, for studies involving humans. Muscle activities were collected as they performed the tasks using surface electromyography (EMG) Trigno Wireless System [19]. Eight EMG sensors were placed on four different muscles of both upper extremities (Biceps Brachii, Triceps Brachii, Extensor Digitorum and Flexor Carpi Radialis). Biceps Brachii was used to bend/flex the elbow and move the forearm close to the body. Triceps Brachii was used to extend the elbow and move the forearm away from the body. Extensor Digitorum was used to extend the wrist whereas Flexor Carpi Radialis was used to bend/flex the wrist. Raw EMG signals were recorded with a sampling rate of 2000 Hz using EMG works Acquisition software [19] and were processed with a band-pass filter of 20–300 Hz and smoothed by a root-mean-square (RMS) technique with a 150-ms moving window to compute the EMG data. To reduce the inter-subject variation, the maximal voluntary contraction (MVC) was obtained from each muscle to normalize EMG signals. All EMG data were presented as the percentage of MVC. Kinematic data such as time, speed, and distance were collected as participants completed each task. In addition, after each task, we surveyed the participants to assess their workloads in terms of mental demand, physical demand, temporal demand, performance, effort, and frustration. The scores of NASA-TLX range from 0 to 10, with 10 being the higher subjective workload.

*2.5. Feature Selection*

We have collected normalized EMG signal data from eight muscles [right extensor digitorum (RED), right biceps (RBC), right flexor carpi radialis (RFCR), right triceps (RTRI), left extensor digitorum (LED), left biceps (LBC), left flexor carpi radialis (LFCR), and left triceps (LTRI)] of 18 subjects over three sessions of work on the NP task to construct network models. To achieve better results, a feature selection method was included to determine the key features affecting a subject's performance.

Stepwise Regression

The term "stepwise regression" refers to a method of iteratively building a regression model in which the independent variables used in the final model are selected at each stage [20]. Each cycle entails either adding or eliminating a potential explanatory variable and then conducting a statistical significance test. Starting with a set of independent variables, the stepwise regression model known as "backward elimination" removes one feature at a time to see if the eliminated variable is statistically significant [20]. In our project, we use the backward elimination method in the stepwise regression model to select the best set of Muscles that impact the Kinematics (time, speed, and distance) of a subject while performing the NP task. The results of stepwise regression are shown in Table 1.

We can observe that among eight muscles, RBC & RFCR are the only muscles that impact the performance of a subject in the NP task. We will build the network models in the next step using the variables (RBC & RFCR for NP task) that have a significant impact on the performance of the subjects.

**Table 1.** Stepwise Regression results showing the subset of variables that impact the needle passing task.

| Coefficients | Estimate | St. Error [1] | t Value [2] | Pr (>|t|) [3] |
|---|---|---|---|---|
| Intercept | −134.924 | 6.3131 | −21.37 | $<2 \times 10^{-16}$ *** |
| RBC | −0.32507 | 0.1357 | −2.396 | 0.0168 * |
| RFCR | 0.25268 | 0.1084 | +2.330 | 0.0200 * |

[1] St. Error: standard error of the regression. [2] t value: estimate/st. error. Measures how many standard errors the coefficient is away from 0. [3] Pr(>|t|): probability of obtaining the t value with significance, significant. Codes: 0 "***", 0.01 "*".

*2.6. Network Model Creation*

The most important part of the analysis is building the network of correlations which is a central component of the proposed comparative population-based approach. Each participant is a vertex (node) in the correlation network graph, and if there is a relationship between any pair of vertices, then an edge connects them. The basis of the correlation graph is the idea that comparing two people's electromyographic (EMG) data might show how similar they are to one another. A strong correlation exists between two subjects' EMG features when they are identical. Conversely, if two people have different EMG characteristics, we call them "weakly correlated" or "distant." In this study, we measured the degree of correlation using Pearson's pairwise correlation coefficient (ρ). Pearson's pair-wise correlation coefficient is a measure of the linear dependence of two data points. Correlation coefficients can take on values between 0 and 1, with 0 indicating no meaningful relationship and 1 indicating an extremely strong one. The flowchart of all the steps involved in network model creation is shown in Figure 2 below.

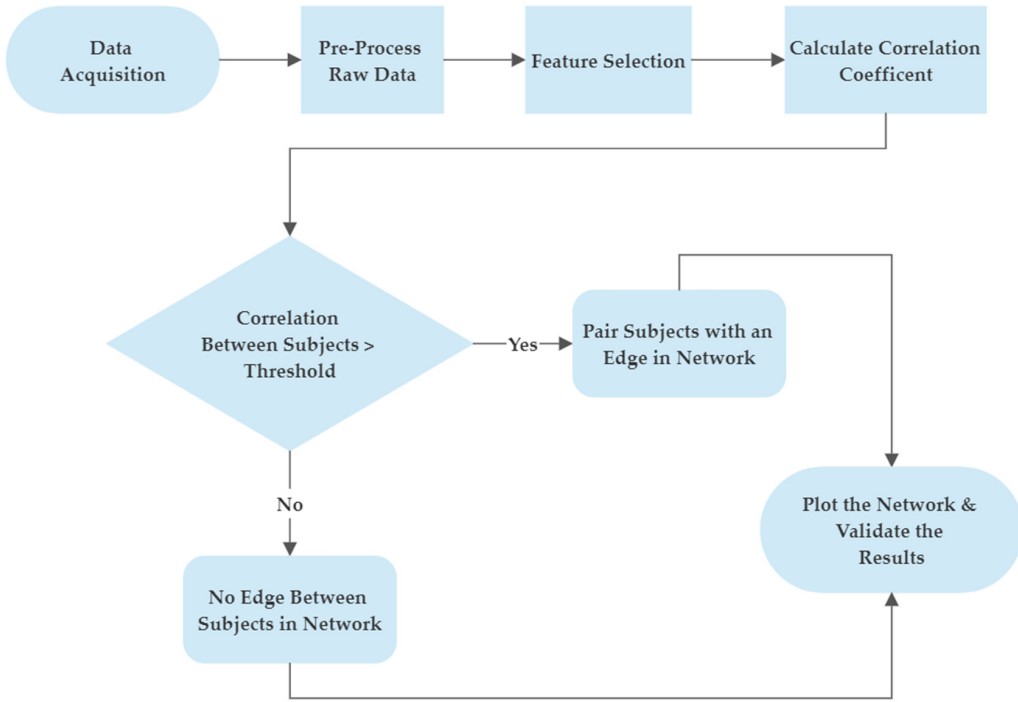

**Figure 2.** Flow chart of all the steps involved in creating a network model.

All the features chosen during the feature selection phase are used to calculate the Pearson pair-wise correlation coefficient for each pair of participants. In this process, we generate a Correlation Matrix (CM) that contains the values of the correlation coefficients between each possible pairs of subjects. Thus, an $18 \times 18$ CM is produced, where a value at CM [i, j] indicates the Pearson correlation coefficient value between Subjects i and j. Subjects i and j are related to one another if the value at CM [i, j] is close to 1, and are not related to

one another if the value at CM [i, j] is close to 0. For example, CM [7,11] has a 0.2 correlation coefficient, indicating a weak association between Subjects 7 and 11, while CM [7,18] has a 0.95 correlation coefficient, indicating a significant relationship between subjects 7 and 18, as shown in Figure 3 in the next section. A ρ value of 0.5 or higher is statistically significant. A correlation criterion "k" is picked to get the groups that are similar enough to be separated. We select a number for "k" that allows us to classify subjects according to their degree of connectivity, with those with close ties together clustered together and those with weaker ties being kept apart. By selecting "k" and solving the corresponding Equation (1), a significance matrix (SM) can be created.

$$SM\,(i,j) = \left\{ \begin{array}{ll} 1, & \text{if } (\rho(Pi, Pj)) \geq k \\ 0, & \textit{For other Cases} \end{array} \right. \tag{1}$$

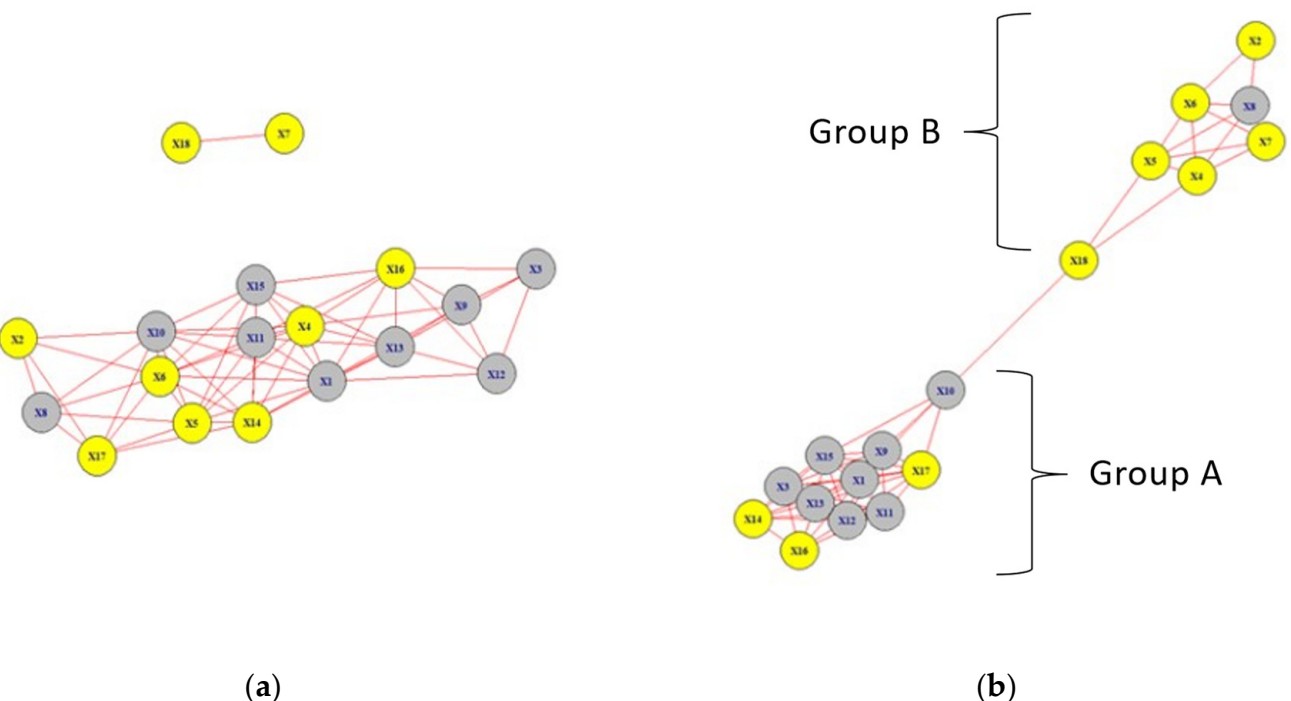

(**a**) (**b**)

**Figure 3.** Network model for the needle passing task: (**a**) network model in Session 1; (**b**) network model in Session 3. Participants who had a medical background are represented by yellow nodes, while participants with a non-medical background are represented by gray nodes. The network formed in Session 3 is divided into two clusters, one with higher concentration of non-medical students (Group A) and other with higher concentration of medical students (Group B).

If SM [i, j] is 1, then the difference in ρ values between i and j is greater than k, and if it is 0, then the difference is smaller than k. In this case, if the correlation value at CM [i, j] is greater than the set threshold (90%), then the resulting SM [i, j] will consist of 1 or 0. The acquired SM is now equivalent to an adjacency matrix, which will represent the 18-node correlation network graph. If SM [i, j] is 1, then any two Subjects i (I ≤ 18) and j (j = 18) are linked together in the network.

### 3. Results

From the data collected, two graphs were created representing participants' overall similarity in performing the needle passing task. The network models for the NP task across two sessions are shown in Figure 3a,b.

### 3.1. Networks Formed in the NP Task

The network formed in Session 1 is quite different from the network formed in Session 3. We can observe that one big cluster in Session 1 has changed to two different clusters in session 3. The two clusters of the network after week 4 are reflecting two groups that have progressed differently after practice. In addition, we can observe that Subjects 7 & 18 are isolated from the only cluster formed in Session 1, but they are connected to each other. The network formed in Session 3 is divided into two clusters, one with higher concentration of non-medical students (Group A) and other with higher concentration of medical students (Group B). There are no isolated nodes in the network formed in Session 3.

### 3.2. Validation of Networks Formed in NP Task

We are validating the Networks constructed in the preceding part by showing the raw distribution of the normalized EMG values used to construct the network. In Figure 4a,b, we can observe the distribution of raw EMG values for participants 7, 18, and the rest of the group performing the NP task in Session 1. As discussed in the Feature Selection, we have only used the EMG values of RBC & RFCR muscles, as they are the only ones that showed an impact on the performance of the subjects. We can see that the distribution of EMG values in Subjects 7 and 18 is comparable but considerably different when compared to the rest of the subjects in the study. As a result, we may say that the networks established in Session 1 for the NP task indicate the true patterns.

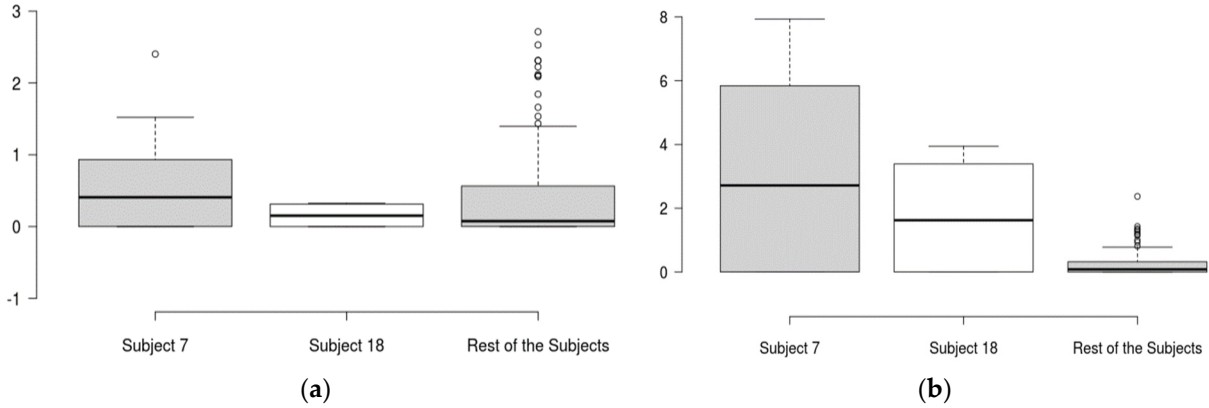

**Figure 4.** Box plot to see the distribution of raw EMG values for NP task in Session 1: (**a**) distribution of RBC muscle values of NP task in Session 1 (**b**) distribution of RFCR muscle values NP task in Session 1.

In Figure 3b, we can observe that Subjects 14, 16, and 17 are medical students with muscle patterns similar to Group A, while Subject 8 is a non-medical student with muscle patterns similar to Group B students. Figure 5a,b demonstrates that the distribution of RBC and RFCR muscle values for Group A is quite comparable to Subjects 14,16,17 and Group B to Subject 8. As a result, we can say that the networks established in Session 3 for the NP task indicate the true patterns.

### 3.3. Enrichment Analysis

The enrichment analysis explains the reasons behind the establishment of groups using parameters that were not used to construct the network. We discovered that the subject's performance can be affected by mental overload levels in our study. As a result, we obtained the NASA TLX scores of the subjects immediately after they completed the exercise. NASA TLX scores comprise the subjects' mental demand (MD), physical demand (PD), temporal demand (TD), performance, effort, and frustration while executing the activity. NASA TLX scores range from 0 to 10, with 10 being the highest subjective effort. Let us look at how the NASA TLX scores differ between groups.

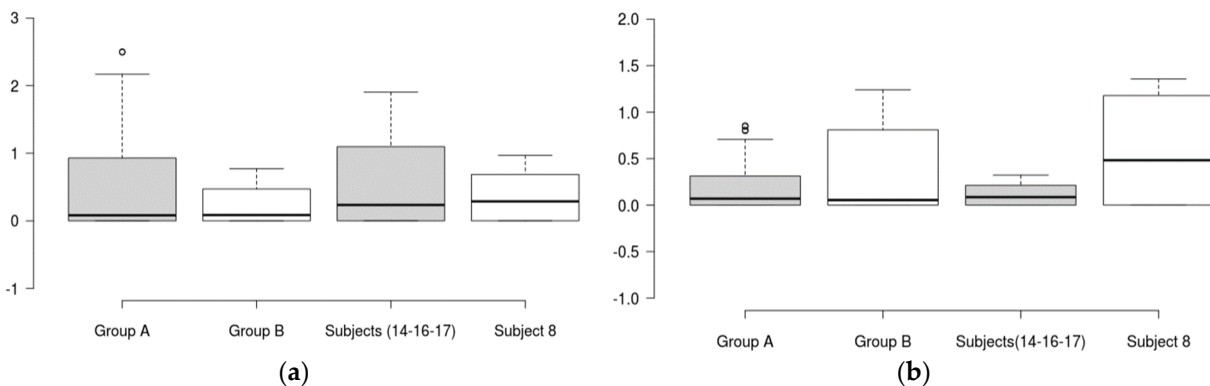

**Figure 5.** Box plot to see the distribution of raw EMG values for NP task in Session 3: (**a**) distribution of RBC muscle values of NP task in Session 3 (**b**) distribution of RFCR muscle values NP task in Session 3.

### 3.3.1. NASA-TLX Scores for NP Task in Session 1

Figure 3a shows that in Session 1, Subjects 7 and 18 are apart from the rest of the group but connected to one another. Subject 7's temporal demand and frustration levels are modest, but they are comparable to other NASA TLX scores, as seen in Table 2. Subject 18 matches the other subjects' scores except that it has the lowest temporal demand and lowest physical demand compared to the other participants. Overall, the NASA TLX results from Session 1 indicate that people exhibit varying levels of frustration and temporal demand when doing the NP task for the first time, despite having comparable muscular motions.

**Table 2.** Nasa-TLX scores for NP task in Session 1.

| Subjects | MD [1] | PD [1] | TD [1] | Performance | Effort | Frustration |
|---|---|---|---|---|---|---|
| 7 | 5.5 | 5.5 | 3.0 | 4.5 | 6.0 | 2.0 |
| 18 | 3.5 | 2.0 | 1.5 | 6.5 | 6.0 | 5.0 |
| Range of remaining Subjects | (2.0–7.5) | (2.5–9.0) | (2.0–10) | (1.0–8.0) | (3.0–9.5) | (0.5–8.5) |

[1] MD: Mental Demand, PD: Physical Demand, TD: Temporal Demand.

### 3.3.2. NASA-TLX Scores for NP Task in Session 3

Figure 3b shows that in Session 3, medical subjects 14,16, and 17 have similar muscle movement patterns among each other and as well as to other non-medical subjects in the study. On the other hand, Subject 8, a non-medical student has similar muscle movement patterns to the other medical subjects in the study. Compared to the group average of NASA TLX scores in Table 3, Subject 8 has higher levels of physical demand (6), mental demand (6), temporal demand (6.5), effort (6) and frustration (6). Subject 14 has a higher level of performance score (8.5) and a lower frustration (0.5). Subject 16 has a greater level of performance (7) and effort (6) but is on par with the remaining NASA TLX scores. Subject 17 has a lower level of performance (2) and a higher level of frustration (7.5) and effort (7) while performing NP tasks in Session 3.

Session 3 NASA TLX data shows that while participants' muscular motions in the NP task are similar, their levels of frustration, effort, performance, and temporal demand vary. The overall picture shows that subjects with extremely high or low scores on at least two of NASA TLX's six dimensions tend to be clustered apart from those who share their background in both sessions.

**Table 3.** Nasa-TLX scores for NP Task in Session 3.

| Subjects | MD [1] | PD [1] | TD [1] | Performance | Effort | Frustration |
|---|---|---|---|---|---|---|
| 8 | 6.0 | 6.0 | 6.5 | 3.0 | 6.0 | 6.0 |
| Range of remaining subjects in Group B | (1.5–5.0) | (0.5–5.5) | (1.5–5.5) | (0.5–6.0) | (2.5–5.5) | (1.5–6.0) |
| 14 | 2.2 | 2.5 | 0.5 | 8.5 | 2.5 | 0.5 |
| 16 | 3.5 | 4.0 | 5.5 | 7.0 | 6.0 | 3.0 |
| 17 | 4.5 | 4.0 | 6.0 | 2.0 | 7.0 | 7.5 |
| Range of remaining subjects in Group A | (1.0–5.0) | (2.0–5.5) | (2.0–7.5) | (1.5–5.5) | (1.5–6.0) | (1.5–6.0) |

[1] MD: mental demand. PD: physical demand. TD: temporal demand.

## 4. Discussion

A major goal of this project was to demonstrate that network models and population analysis can be employed successfully in creating an automatic and objective tool for assessing the performance of participants in using new training systems or medical devices. Our results show that the network model implemented over time, for three sessions, is successful in showing the progression of the participants as it has been shown to be successful in other application domains [21–23]. In the research on laparoscopic skills assessment, individual performance-based evaluation is widely used [24–27]. Most studies usually evaluate differences between the groups in terms of measurement average values (e.g., mean of the kinematic measures) without assessing the change in individual performance [25]. Applying network models to assess skills performance not only can detect individual differences across time, but also associate the individual changes with the whole group as well as assessing different training schedules as done by Rastegari et al. [23].

The work published by Rastegari et al. [23], is the closet study to the present work in which they applied network analysis using features extracted from graspers' motions while performing simulated surgical skill. Their results showed that students' progress in performing a task is quicker if the training sessions are held more closely compared to when the training sessions are far apart in time. They also reported that using a network model takes the objective assessment of individuals' surgical proficiency level one step further. Our results are consistent with these findings, indicating that using network models on the data collected over training sessions can reveal a network pattern in which participants with similar level of proficiency are connected. Moreover, in this paper we applied enrichment analysis and used NASA TLX measures to explain the network models. This is the first time that enrichment analysis and NASSA TLX measures are used in assessing how simulated surgical tasks are performed and if there is any correlation between factors such as mental demand or perceived frustration of the participants on how they perform these tasks. The network model generated using the data collected from the first session of training is a homogeneous network in which most of the subjects are connected, except two of them. Having a homogeneous network in the beginning makes sense because the expertise level of everyone in terms of performing the needle passing task is similar. This is consistent with the result of another study using network models for evaluating the expertise level of participants over time learning how to perform simulated peg transfer and color matching tasks [23]. However, network models generated for a cube transfer task in the same study [23] does not follow the same pattern and we do not see a homogeneous network in the beginning. This could be due to the nature of the tasks, how easy or difficult they are, and what sorts of skills they need to be completed.

The network model generated using the third session's data shows two clusters of participants; one that mainly consists of medical students and one that mainly includes non-medical students. This indicates that the way and the speed in which these two groups developed the required skills to perform the needle passing task is different.

The analysis of NASA TLX data shows that each of the NASA TLX factors weakly correlate with muscular motions. However, it seems that a combination of NASA TLX factors could be considered for evaluation of the network model and muscular motion

patterns. Looking at the first session's network model and two individuals that are isolated from the rest but connected to each other, we can see that both are medical students, both with relatively low temporal demand, one with low level of frustration and the other one with the lowest physical demand. The reason why these two participants are connected in the network model and have similar muscular motions could be because they perceive the task similarly in terms of NASA TLX overall rating. Coming up with one measure as NASA TLX overall rating using its factors and then applying it to get more insight from the network model could be of future research.

In the study by Lefor et al. [28], robot-assisted surgery kinematic data was used to develop predictive models of skill required for three tasks (suturing, knot-tying, and needle passing). They analyzed the relationships of self-defined skill level with global rating scale scores and kinematic data. In our study we only used kinematic measures to find out the muscles' features that correlate the most with them, but the correlation of kinematic features with each factor of NASA TLX was not studied. This could have guided us in picking certain NASA TLX factor/factors for gaining more insight about the network models. Additionally, the idea of comparing self-defined skills with expert-defined and rated skills using network models could be of our future studies.

Such network-based assessment provides a multi-dimensional view for clinical education program directors in medical school to monitor the learning progression of each medical student or fellow, compare a certain trainee within a cohort, and examine between-trainees performance in a training cohort across time. As a result, program directors can develop an optimal and individualized learning environment for trainees to efficiently acquire surgical skills performance. This relative assessment approach also allows trainers and trainees to monitor their progress over the duration of the training period. One of the most benefits of the proposed approach is the ability to customize the training for each participant as well as for each skill set, particularly when it comes to deciding on a follow-up training session for the trainees.

*Limitations*

The main limitation of this study is the relatively small sample size. Additional studies with larger sample sizes are needed to generalize the reported findings. In addition, the temporal analysis could benefit from having additional data points for each participant. We hope that the dissemination of the results of our study will encourage more groups conducting training workshops to make their data available for similar analysis. The availability of additional training datasets would make it possible to conduct the population analysis using network models at multiple levels and obtain valuable insights into how to provide objective assessments of education and training exercises in various domains.

## 5. Conclusions

Analysis of the population's progression could be useful in an environment where trainees must be efficiently trained to use simulation training. While this study specifically examined how a network model could be used to analyze the performance of diverse groups of students or trainees in basic tasks essential to laparoscopic surgery, this analytical approach can certainly be used in a wide range of training applications such as in the areas of robotic surgery, ophthalmology, obstetrics, and gynecology. The next step of this study will focus on incorporating additional data to further test the proposed approach and assess its impact on analyzing different types of training data. This step will utilize data from medical training partners as well as data collected from publicly available data.

**Author Contributions:** Conceptualization, H.H.A. and S.M.; methodology, S.M.; software, S.M.; validation, E.R., H.H.A. and K.-C.S.; formal analysis, S.M.; investigation, S.M.; resources, K.-C.S.; data curation, K.-C.S.; writing—original draft preparation, S.M.; writing—review and editing, K.-C.S., H.H.A. and E.R.; visualization, S.M.; supervision, H.H.A.; project administration, H.H.A. and K.-C.S.; funding acquisition, K.-C.S. All authors have read and agreed to the published version of the manuscript.

**Funding:** This research was partly funded by the NASA Nebraska Space Grant (NNX10AN62H).

**Institutional Review Board Statement:** The study was conducted in accordance with the Declaration of Helsinki and approved by the Institutional Review Board of the University of Nebraska Medical Center (IRB # 103-12-EX) on 11 January 2017, for studies involving humans.

**Informed Consent Statement:** Informed consent was obtained from all subjects involved in the study. Written informed consent has been obtained from all subjects to publish this paper.

**Data Availability Statement:** Data is contained within the article.

**Conflicts of Interest:** The authors declare no conflict of interest. The funders had no role in the design of the study; in the collection, analyses, or interpretation of data; in the writing of the manuscript; or in the decision to publish the results.

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
