# Peer review of "An Innovative Comparative Analysis Approach for the Assessment of Laparoscopic Surgical Skills"

_2673-4095, doi:10.3390/surgeries4010007_

Round 1

Reviewer 1 Report

I read with interest the manuscript entitled "An innovative comparative analysis approach for the Assessment of Laparoscopic Surgical Skills"

Below are some recommendations for improving the manuscript;

In the introduction, you mention that there is a growing shortage of medical professionals in the United States. Please also mention the rest of the world, considering that the mentioned problem does not exist only in the USA. I suggest that you present concrete facts about the reduction of medical professionals in the last decade, especially surgical ones.

Please state more clearly the primary and secondary objectives of your study at the end of the introduction.

Do you think a sample of 18 individuals divided into two groups is sufficient to draw conclusions?

How did you choose 18 individuals? Are there any limitations that should have been taken into account? Does everyone meet the criteria to be included in the study? Please indicate the inclusion and exclusion criteria.

Figure 1 is of extremely poor quality. Please replace it with a better one.

Under the table, write the meaning of the abbreviations.

I suggest that you briefly note which movements are performed by which muscle. Please add to additional materials.

Considering the sample, did you use Pearson's correlation coefficient justified?

How did you interpret the correlation of 0.5?

Do you think your study has no limitations? Please specify the limitations.

You yourself are aware of the fact that every surgical procedure, in reality, has its specificities, and within the procedure itself there are individual problems for each patient. Do you think that with this model you can recruit the best individuals for each specific situation that a surgeon may encounter in reality?

Please shorten the conclusion. The part can certainly be incorporated into the discussion.

Reviewer 2 Report

A very well-written study. My only suggestion would be to reduce the length of the Conclusion by adding some of the content to the Discussion instead. Moreover, the limitations of this study, as well as the limitations of using the described analysis method, must be discussed.

Round 2

Reviewer 1 Report

There is no doubt that you have made the manuscript better.

Start the discussion by presenting your main results, which you then compare with other studies. Do not repeat the objectives and methods in the discussion!

I also recommend that you start the conclusion section with the sentence "Analysis of the population's progression...". Please remove the first sentence from the conclusion.
